# *Cinnamomum zeylanicum* Blume Essential Oil Inhibits Metastatic Melanoma Cell Proliferation by Triggering an Incomplete Tumour Cell Stress Response

**DOI:** 10.3390/ijms24065698

**Published:** 2023-03-16

**Authors:** Giulia Cappelli, Daniela Giovannini, Laura Vilardo, Annalisa Basso, Ilaria Iannetti, Marianna Massa, Giuseppe Ruberto, Ryan Muir, Carlo Pastore, Igea D’Agnano, Francesca Mariani

**Affiliations:** 1Institute for Biological Systems (ISB)-CNR, Via Salaria Km 29, 00015 Monterotondo, Italy; 2Institute of Biochemistry and Cell Biology (IBBC)-CNR, Via E. Ramarini 32, 00015 Monterotondo, Italy; 3Institute for Biomedical Technologies (ITB)-CNR, Via Fratelli Cervi 93, 20054 Segrate, Italy; 4Institute for Biochemical Chemistry (ICB)-CNR, Via Paolo Gaifami, 18, 95126 Catania, Italy; 5Department of Pharmaceutical Chemistry, University of California, UCSF Byers Hall MC2552, San Francisco, CA 94158, USA; 6Sanatrix Clinic, Via di Trasone 61, 00199 Rome, Italy

**Keywords:** antioxidant response, botanicals, *C. zeylanicum*, ferrous iron, human melanoma

## Abstract

Given the known pro-oxidant status of tumour cells, the development of anti-proliferative strategies focuses on products with both anti- and pro-oxidant properties that can enhance antitumour drug cytotoxicity. We used a *C. zeylanicum* essential oil (CINN-EO) and assessed its effect on a human metastatic melanoma cell line (M14). Human PBMCs and MDMs from healthy donors were used as normal control cells. CINN-EO induced cell growth inhibition, cell cycle perturbation, ROS and Fe(II) increases, and mitochondrial membrane depolarization. To assess whether CINN-EO could affect the stress response, we analysed iron metabolism and stress response gene expression. CINN-EO increased HMOX1, FTH1, SLC7A11, DGKK, and GSR expression but repressed OXR1, SOD3, Tf, and TfR1 expression. HMOX1, Fe(II), and ROS increases are associated with ferroptosis, which can be reversed by SnPPIX, an HMOX1 inhibitor. Indeed, our data demonstrated that SnPPIX significantly attenuated the inhibition of cell proliferation, suggesting that the inhibition of cell proliferation induced by CINN-EO could be related to ferroptosis. Concurrent treatment with CINN-EO enhanced the anti-melanoma effect of two conventional antineoplastic drugs: the mitochondria-targeting tamoxifen and the anti-BRAF dabrafenib. We demonstrate that CINN-EO-mediated induction of an incomplete stress response specifically in cancer cells affects the proliferation of melanoma cells and can enhance drug cytotoxicity.

## 1. Introduction

Malignant melanoma is a neoplasm arising in melanocytes, and its incidence has continually increased over the past decades [1]. Although surgery remains a definitive treatment for primary cutaneous melanoma, it is not curative for metastatic melanoma, and its prognosis is generally poor, with a mean survival of 6–8 months; thus, this cancer represents a major cause of morbidity and mortality. The poor prognosis of advanced stage melanoma is in part due to the failure of the therapeutic options available; therefore, novel therapeutic approaches are needed [1].

Oxidative stress is a redox imbalance caused by an increase in reactive oxygen species (ROS) inside the cells [2].

There is increasing evidence that oxidative stress leads to detrimental biochemical reactions and is an important pathogenic factor in several chronic human diseases, including atherosclerosis, neurodegeneration, immunologic disorders, cancer, and even the ageing process [3,4].

In particular, healthy epidermal melanocytes of the skin are highly susceptible to oxidative stress due to the ROS production occurring during melanin biosynthesis [2].

The crucial role played by the pro-oxidant status of tumour cells compared with healthy cells is now generally recognised, and the development of anti-proliferative tools has included a focus on natural products with both anti- and pro-oxidant properties that have been suggested to enhance the cytotoxicity induced in malignant tumour cells [5].

Recent research has revealed that certain natural products, such as terpenes, flavonoids, and anthocyanins, have powerful abilities to counteract ROS accumulation in tissues by inhibiting free radical cascades [6]. So far, different botanicals and essential oils (EOs) have been studied for their anti-proliferative properties against tumour cells [7]. In particular, different purified components of Cinnamomum spp. (such as *Cinnamomum cassia* and *Cinnamomum zeylanicum*), as well as other plant extracts, have been demonstrated to affect human melanoma proliferation [8,9,10]. However, the related studies have been performed by testing single components purified from cinnamon in murine models of melanoma [11].

Of note, the majority of plant polyphenols display the capacity to act as anti- or pro-oxidant substances according to the intracellular milieu upon administration and to exert protective effects in adverse situations such as inflammation, CVD, ageing, and oxidative bursts [12,13,14].

The intracellular redox state is tightly regulated by the labile iron pool (LIP) available in the cell, which is represented by metabolically active ferrous iron, Fe(II). Recently, by using a reactivity-based probe of the intracellular LIP, researchers have demonstrated that LIPs are larger in cancer cells than in non-tumorigenic cells [15]. Accordingly, recent clinical studies have shown that serum iron biomarkers and dietary iron are associated with tumour incidence [16].

On the other hand, iron is also responsible for a recently identified type of programmed non-apoptotic cell death called ferroptosis, which is triggered by augmented LIPs in cells [17,18]. Such a mechanism could actually be exploited to selectively kill tumour cells, which are richer in Fe(II) content than normal cells. A gene that has been recently associated with ferroptotic cell death is Heme Oxygenase 1 (HMOX1), whose inhibition by its specific inhibitor, tin protoporphyrin IX (SnPPIX), completely reverses ferroptosis [19]. In addition, p53 also contributes to tumour suppression through regulation of cystine metabolism, reactive oxygen species responses, and ferroptosis, and these effects highlight the crucial role of SLC7A11, which encodes a component of the cystine/glutamate antiporter [20].

We hypothesised that *Cinnamomum zeylanicum* EO, due to its composition being extremely rich in reactive phytochemicals, could exert a pro-oxidant effect on tumour cells, which are already in a pro-oxidised state. Therefore, we tested the effect of this EO on human melanoma cells, which are highly susceptible to oxidative stress. The main objective of our study was to determine whether the *C. zeylanicum* EO could influence the growth of melanoma cells alone or in combination with conventional chemotherapy agents. The idea would be to use *C. zeylanicum* EO in integrative medicine in combination with conventional antitumor treatments.

## 2. Results

### 2.1. Cinnamomum zeylanicum Whole Essential Oil Composition

Appendix A lists the composition of the essential oils from six commercial samples of cinnamon (Table 1).

In total, 90 components were fully identified. For easier comparison of the oils, these were grouped into four classes: monoterpene hydrocarbons, oxygenated monoterpenes, sesquiterpenes, and others (non-terpenoid components), as shown in Figure 1.

All oils showed a low content of terpenoid and a high content of non-terpenoid components (Appendix A), namely, phenyl-propanoid (C6-C3) derivatives, which are typical characteristics of cinnamon essential oils. Figure 1 shows the molecular formulas of the most significant components. The main component of five of the samples (except for EO4) was E-cinnamaldehyde, ranging from 55 to 87%; sample EO4, instead, showed eugenol as the main component (76%). The total amount of terpenes ranged from 1.1% (EO3) to 38.78% (EO6). The data obtained in this study are in accordance with literature reports on this species; E-cinnamaldehyde and its derivatives are in fact present in very high concentrations in the bark of *C. zeylanicum* [21,22,23,24], whereas high concentrations of eugenol are present in essential oils obtained from the leaves [25].

In particular, sample EO3 presented a peculiar composition quite different from that of all the other essential oils in this study, with E-cinnamaldehyde as the main component (87.04%) and o-methoxy-cinnamaldehyde as the second most abundant component (7.13%); the fractions of all the other components were below 1%. Among the other oils, EO1 and EO2, which have the same commercial origin, showed very similar profiles (Appendix A); in fact, E-cinnamaldehyde was the main component, with percentages of 55.26 and 53.04, respectively; the second component was, in both cases, eugenol with percentages of 21.35 and 21.69, respectively. Similarities were also found in the terpenoid fractions: linalool and β-caryophyllene reached amounts very close to 4% in the two samples, whereas the fractions of the monoterpene hydrocarbons α-pinene, α- and β-phellandrene, and *p*-cymene were more than 1%. The EO4 was characterised by very low levels of cinnamyl derivatives. Eugenol was the main component at approximately 76%, and benzyl benzoate and eugenol acetate were the other two main non-terpenoid components with 3.3 and 2.1%, respectively; among the terpenes, linalool and β-caryophyllene, with ca. 2.6 and 2.3%, respectively, were the main components. The different and particular composition of this oil, with respect to the compositions of the other ones in this study, suggests that this commercial oil was obtained through distillation of the leaves of the plant. Finally, it is useful to highlight the differences between the last two oils (EO5 and EO6) and the previous four based on the classes of the components. In this case, it is observed that the most evident difference is that, in comparison with the first four EOs, the two last oils have a higher concentration of all terpenoidic components (mono- and sesquiterpenes; EO5 37.19%, EO6 38.78%) compared with the non-terpenodic components (others; EO5 62.62%, EO6 60.82%).

### 2.2. C. zeylanicum Essential Oil Cytotoxic Effects on the Human Melanoma Cell Line M14

To determine whether *C. zeylanicum* (CINN-EO) elicits a cytotoxic effect on M14 melanoma cells, we performed dose–response experiments by continuously exposing the cells to different concentrations of CINN-EOs for up to 72 h of cell growth (48 h of treatment), after which we selected the most cytotoxic oil, EO6 (Figure 2).

All subsequent experiments were then performed with EO6, hereafter referred to as CINN-EO. The effect of the chosen CINN-EO was also verified on healthy human cells as a control. We used human PBMCs and MDMs obtained from healthy donors. In both cell types, CINN-EO did not elicit any significant effect on cell viability (Appendix A).

The dose–response curves in Figure 3A clearly show that CINN-EO inhibited M14 cell growth in a dose-dependent manner. Based on these results, we chose a CINN-EO dose of 10 µg/mL, which elicited the maximum inhibitory effect on cell proliferation. As shown in Figure 3B, the inhibition of cell growth was associated with a significant accumulation of cells (74%) in the G2/M cell cycle phases after 24 h of exposure, with a concomitant depletion of cells in the G0/G1 and S phases. This effect was reversible, as the cells appeared to re-enter the cell cycle after 48 h of exposure to the EO. However, when we re-administered CINN-EO, after another 24 h, all cells detached and died.

### 2.3. CINN-EO Affects Intracellular Oxidative Properties and Iron Metabolism

Being aware of both the pro- and antioxidant properties of many essential oils, we investigated the effect of CINN-EO on ROS production in M14 cells and normal hu-PBMCs. As shown in Figure 4, we observed opposite effects of stimulation on these cellular populations.

Exposure of the melanoma cells to CINN-EO for 24 h significantly increased ROS; CINN-EO functioned as a pro-oxidant, generating ROS (Figure 4A, right). In contrast, CINN-EO did not induce any ROS production when administered to normal hu-PBMCs (Figure 4A, left). Consistent with the increase in ROS production, mitochondrial membrane depolarization was observed in M14 cells after treatment with CINN-EO (Figure 4B, right). As expected, CINN-EO did not significantly affect the mitochondrial membrane potential in normal hu-PBMCs (Figure 4B, left).

Using a reactivity-based probe of the intracellular labile Fe(II) pool, Trx-Puro conjugate 3, we measured the Fe(II) pool variations in M14 cells upon CINN-EO treatment. Since Trx-Puro conjugate 3 was detected with an anti-puromycin antibody, we ascertained we were in fact detecting puromycin within the cells. We exposed the cells to 1 µM puromycin for 4 h and then performed immunofluorescence detection as specified in the Materials and Methods. Appendix A shows representative FACS cytograms showing that more than 90% of the cell population was positive for puromycin incorporation. The FACS cytograms reported in Figure 4C clearly demonstrate that CINN-EO treatment increased the M14 intracellular Fe(II) content (the MFI was 44.60 in control cells vs 62.35 in CINN-EO treated cells). However, given that puromycin is an inhibitor of protein synthesis, to better demonstrate the effect of CINN-EO on the Fe(II) pool, we normalized the MFI values of the control and treated cells to their respective protein contents. As shown in Figure 4D, CINN-EO treatment increased the intracellular Fe(II) content by approximately 30% compared with no treatment. Human PBMCs were subjected to the same treatment without showing a significant Fe(II) pool increase.

The observed modulation of ROS production and intracellular Fe(II) pools prompted us to analyse the expression level of a group of genes involved in regulating the antioxidant response and intracellular iron. For iron metabolism, we measured, by qPCR, SLC40A1 (or Ferroportin, Fp), SLC11A1 (or Nramp1), Transferrin Receptor 1 (TfR1), Heme Oxygenase 1 (HMOX1), and Ferritin Heavy chain (FTH1), both in M14 cells and human MDMs. As shown in Figure 4E, both Fp and Nramp1 were undetectable in M14 cells, and their levels in control MDMs were not significantly modified by CINN-EO. TfR1 mRNA was found in M14 cells, but its levels were not significantly modulated upon CINN-EO treatment (TfR1 was stable in MDMs, as expected). The last two iron-regulating genes, HMOX1 and FTH1, were instead significantly induced in CINN-EO-treated vs. untreated M14 cells. The same gene expression induction was found for the Diacylglycerol Kinase kappa (DGKκ) and Glutathione-Disulfide Reductase (GSR) antioxidant genes, while inducible Nitric Oxide Synthase (NOS2) and Superoxide Dismutase 2 (SOD2) did not change significantly; SOD3 showed a repression trend upon CINN-EO treatment, but the change was not statistically significant.

### 2.4. Effects of CINN-EO and Tamoxifen Co-Administration on M14 Cell Survival, Gene Expression and Protein Level

The drug tamoxifen (TAM) is now under reconsideration for possible use in melanoma therapy; therefore, given that CINN-EO can act as a pro-oxidant and that TAM can affect mitochondrial integrity [26,27], we determined whether combination treatment could potentiate the cytotoxicity produced by the two compounds when given as single agents. We chose a TAM dose of 0.1 µM, which does not produce any cytotoxic effect on M14 cells. As shown in Figure 5A, co-administration of CINN-EO and TAM significantly reduced M14 cell survival, not only vs. no treatment (*p* < 0.0001) but also vs. CINN-EO alone (*p* = 0.0009) and vs. TAM alone (*p* < 0.001). Figure 5B shows representative images of the colonies formed after the different treatments. Interestingly, as shown in Figure 5C, CINN-EO treatment in combination with TAM modified the expression profiles of only those genes associated with the cell antioxidant response, such as HMOX1, FTH1, DGKK, GSR, SLC7A11 (induced by TAM+CINN-EO), and OXR1 (repressed by TAM+CINN-EO), compared with CINN-EO treatment alone.

We wondered whether CINN-EO alone or in combination with TAM might have changed the expression of cell cycle and stress-related proteins, such as Cyclin B1, HSP72/73 (involved in ER stress), p53, and the iron-responsive protein HMOX1. As expected, Cyclin B1 levels increased upon treatment with both CINN-EO and TAM+CINN-EO, confirming the arrest of cells in the G2/M cell cycle phases. In addition, the levels of HSP72/73 and p53 increased after CINN-EO treatment, and co-administration of the oil with TAM significantly enhanced this effect, further increasing the HSP72/73 and p53 levels by approximately 3-fold and approximately 1.5-fold, respectively (Figure 6A,B). The expression of HMOX1, which was present after CINN-EO and TAM+CINN-EO treatment (Figure 6C), was absent in untreated and TAM-treated M14 melanoma cells.

### 2.5. Effects of CINN-EO and Paclitaxel or Dabrafenib Co-Administration on M14 Cell Survival

We also studied the effect of CINN-EO in combination with other two conventional drugs used in melanoma therapeutics, the antimitotic paclitaxel (PTX) and the BRAF inhibitor dabrafenib (DAB), on melanoma cell survival.

*C. zeylanicum* EO did not enhance the effect of PTX on M14 colony-forming ability (CFA). Administration of PTX alone elicited an inhibitory effect on M14 CFA of about 10%, whereas administration of CINN-EO showed an inhibitory effect on M14 cell survival of 45% (*p* = 0.0002). This effect was not increased when the two agents were administered in combination, the inhibitory effect on M14 cell survival being about 40% after PTX+CINN-EO treatment (Figure 7A). Figure 7B shows representative images of the colonies formed after the different treatments.

CINN-EO given in combination with DAB to M14 cells was able to significantly enhance the effect of two different doses of DAB used alone on M14 CFA. Based on the dose/response effect of DAB on cell viability, we chose the low 1 nM and the highest 1 µM doses of DAB (Appendix A). In both combinations, CINN-EO significantly enhanced the effect of DAB used alone (*p* = 0.02) (Figure 7C). Figure 7D shows representative images of the colonies formed after the different treatments.

### 2.6. Effect of an HMOX1 Inhibitor on CINN-EO-induced Cytotoxicity

Given the crucial role played by HMOX1 in ferroptosis and the displayed gene upregulation upon CINN-EO treatment, we wanted to ascertain the importance of the HMOX1 protein in the anti-proliferative effect observed in treated melanoma cells. The data shown in Figure 8 confirm that CINN-EO-induced cytotoxicity was associated with the full activity of the HMOX1 protein. Inactivation of HMOX1 by administration of the HMOX1 inhibitor SnPPIX allowed, in fact, a significant and complete recovery of melanoma cell proliferation and metabolic activity.

## 3. Discussion

In the present study, we demonstrate, for the first time, that *C. zeylanicum* essential oil (CINN-EO) inhibits cell proliferation in the M14 human metastatic melanoma cell line. This effect was evidenced by the accumulation of cells in the G2/M cell cycle phase and increases in ROS production. We also found that CINN-EO augmented the Fe(II) content within tumour cells and induced HMOX1 expression, suggesting that ferroptotic cancer cell death was activated. To assess whether CINN-EO might be detrimental to human immune cells, we used as control cells primary cultures of monocyte-derived-macrophages (MDMs) and human peripheral blood mononuclear cells (hu-PBMCs), which were not affected by CINN-EO treatments. In addition, we also found that co-administration of CINN-EO and two different antineoplastic drugs (tamoxifen and dabrafenib) resulted in the enhancement of the antitumour effect produced by the chemotherapy agent used alone. Our data clearly suggest a possible use of CINN-EO in the management of human melanoma.

It is now widely recognised that tumour cells are in a pro-oxidant redox state. Although increasing ROS generation has recently been considered a valuable method by which to kill tumour cells, this might be insufficient due to cancer cell adaptation strategies. The use of phytochemicals in combination with chemotherapy has highlighted the capacity of these natural substances to further increase ROS in the tumour milieu and affect tumour cell redox homeostasis [28].

Melanocyte transformation into cancer cells is associated with significant structural alterations in melanosomes. In addition to producing pigment, melanosomes also protect cells by scavenging free radicals generated by sunlight and cellular metabolism. In melanoma, disruption and disorganization of the melanosome structure reverses this process. Melanosomes found in melanoma produce free radicals, such as hydrogen peroxide, exacerbating DNA damage [29].

In our study, we employed *C. zeylanicum* essential oil (CINN-EO), and we observed CINN-EO-mediated cytotoxicity against M14 metastatic melanoma cells that was associated with significant increases in ROS production. This observation leads us to think that the increased ROS production might have contributed to the inhibition of cell growth observed in M14 cells following CINN-EO exposure.

Consistent with our data are results obtained by Tuma et al., who described a clinical trial including metastatic melanoma patients treated with paclitaxel plus the copper-chelating agent elesclomol, a ROS-inducing drug. Patients taking elesclomol had a median progression-free survival of 3.7 months compared with 1.8 months for those in the control arm—a statistically significant difference [30].

However, the cytotoxic effect observed in our study was reversible, as evident from the cell regrowth and re-entry into the cell cycle after 48 h of CINN-EO exposure. Further administration of CINN-EO every 24 h to M14 cells led to definitive arrest of the cells in the cell cycle and subsequent cell death.

The increased ROS production was associated with significant increases in Fe(II) levels and in HMOX1 gene and protein expression, suggesting that a ferroptosis programmed cell death pathway could have been engaged. HMOX1 plays a central role in CINN-EO-mediated cytotoxicity, as demonstrated by the significant reversal of the inhibitory effect induced by CINN-EO on M14 cell proliferation after treatment of the cells with the HMOX1 inhibitor SnPPIX.

The role of HMOX in the fate of tumour cells is contradictory; it has been described as either cytoprotective or ferroptosis-promoting. In fact, HMOX is likely to switch from one role to the other depending on the oxidative states of cells [31].

The pattern of expression of this stress response protein has also been quite variable in different human melanoma studies [32,33,34], but it is worth noting that in our study, the HMOX1 protein pattern of expression was induced exclusively by CINN-EO, being absent in untreated metastatic melanoma cells.

We therefore wondered whether CINN-EO co-treatment might enhance M14 melanoma cell sensitivity to three different antineoplastic agents. Tamoxifen (TAM), to which the cells normally display resistance, and which is now under reconsideration for melanoma chemotherapy [35]; paclitaxel (PTX); and dabrafenib (DAB). We showed that co-administration of CINN-EO and TAM or DAB had an effect superior to that of the two individual treatments on M14 survival and could allow the use of the drugs at doses lower than the ones employed in current protocols.

Moreover, treatment with CINN-EO alone and co-treatment with CINN-EO and TAM both modulated the genes HMOX1, DGKK, and SLC7A11, which increased, and OXR1, which decreased.

At the protein level, the effects on the expression of HMOX1 induced by CINN-EO alone and by CINN-EO and TAM co-treatment support the possible role of HMOX1 in M14 ferroptotic cell death, while the effects on p53, HSP 72/73 and Cyclin B1, whose expression was higher after co-treatment than after single-agent treatment, were reasonable indicators of ER stress and cell cycle arrest.

Another gene highly induced by CINN-EO treatment was DGKκ. Due to the ability of DGK to convert one important signalling molecule diacyl glycerol (DAG) into another, namely, phosphatidic acid (PA), the activity of DGK in different cells is tightly controlled for the maintenance of normal physiological conditions. In mammalian species, in which the DGK family of proteins has been best studied, 10 different isozymes of DGK differing in their biochemical properties, tissue distributions, and lengths have been identified [36].

A genome-wide association study also indicated a potential relationship between DGKκ and hypospadias, a common congenital malformation of the male external genitalia [37]. Several reports have revealed that DGKδ, η, and κ are abundantly expressed in reproductive organs, including the testis and ovary.

Intriguingly, the DGKκ isoform has never been found in tumour tissues, while we found it to be expressed in CINN-EO-treated melanoma cells and in melanoma tissue from patients in gene repositories, albeit at low levels (TCGA). Interestingly, in the human testis, different MAGE (melanoma antigen) genes are also expressed, which are known to regulate androgen binding to Androgen R and are also found in human melanoma. Notably, it has been suggested that this isozyme changes the balance of signalling lipids in the plasma membrane in response to oxidative stress [38]; this function might explain the increased DGKκ expression induced by CINN-EO as a consequence of the augmented ROS in melanoma cells. What might appear controversial is that DGKκ, but not other type II DGK isozymes, has been shown to be specifically tyrosine-phosphorylated and downregulated in H2O2-treated COS-7 cells [38]. Most likely, the African green monkey kidney fibroblast cell line COS-7 displays an initial intracellular ROS burden different from that in M14 melanoma cells with a pro-oxidant status.

SLC7A11, another CINN-EO-induced gene (also known as xCT), is the light chain subunit of the cystine/glutamate antiporter system Xc−. It plays a vital role in maintaining cell redox homeostasis and has been shown to be upregulated in a compensatory manner by Xc- inhibitors such as erastin [39].

Cystine uptake across the cell membrane helps tumour cells to increase intracellular glutathione biosynthesis; SLC7A11-mediated glutamate export limits intracellular glutamate (Glu) supply to the TCA cycle and mitochondrial respiration, rendering cells more dependent on glucose and/or glutamine supply for survival and growth. In addition, elevation of extracellular glutamate levels creates excitotoxicity and facilitates inflammation in the CNS [40]. Regulator of G protein signalling 4 (RGS4) has been found to inhibit human melanoma (in the same cell line tested in this study, M14), and RGS4-silenced neuroblastoma exhibits decreased SLC7A11 expression [41], suggesting a possible role of RGS4-mediated increase in SLC7A11 in the course of melanoma inhibition.

In our study, the antioxidant response of M14 melanoma cells to CINN-EO was imperfect, as evidenced by the decreases in OXR1 and SOD3 gene expression. The OXR1 gene was repressed by CINN-EO. This gene has been shown to be able to prevent intracellular hydrogen peroxide-induced increases in oxidative stress levels and to prevent the vicious cycle of increased oxidative mitochondrial DNA (mtDNA) damage and ROS formation [42]. OXR1 has also been characterised as an important antioxidant protein that regulates the expression of a variety of antioxidant enzymes in senescent cells (SCs), and it has been found to be a target of senolytic agents, which selectively deplete SCs as a therapeutic approach for treating age-related diseases [43]. Its repression might therefore have reduced M14 melanoma cell resistance to CINN-EO-induced ROS increases. The SOD3 gene was also repressed by CINN-EO treatment in M14 melanoma cells. Notably, SOD3 has been found to be expressed throughout the epidermis and dermis, and its levels are altered upon the progression of inflammation. The SOD3-mediated defence mechanism responds to superoxide, particularly when stimuli or environmental factors affect tissues or cells where SOD3 is specifically expressed. The expression of SOD3 in the dermis may be related to the protection of matrix components against superoxide, which may be effective in preventing skin ageing and cancer [44]. Therefore, SOD3 repression upon CINN-EO treatment might have exacerbated M14 melanoma cell resistance to the increased superoxide burden.

In a recent study, the authors identified a homologue of human OXR1 (LMD-3) in the nematode *Caenorhabditis elegans* (*C. elegans*). Their results indicate that, in cooperation with mitochondrial SODs (SOD2 and SOD3), LMD-3 contributes to protection against oxidative stress and ageing in *C. elegans* [45]. This functional association between OXR1 and SOD3 supports the possible detrimental role played by the CINN-EO-induced repression of both these genes in M14 cells in response to augmented ROS and Fe(II).

Finally, one study has documented an immunosuppressive effect of cinnamaldehyde on a human monocytic cell line and on hu-PBMCs due to inhibition of proliferation and induction of apoptosis [46]. It is important to keep in mind that the hu-PBMCs were treated with a purified compound derived from cinnamon and not with a whole and balanced phytocomplex. Nevertheless, in our experiments, CINN-EO did not display cytotoxic effects towards human MDMs.

## 4. Materials and Methods

### 4.1. Cinnamomum zeylanicum Essential Oils

The six essential oils (EO1–EO6) used in this study are listed in Table 1, all of which were commercial samples. Each analysis was repeated in triplicate. All the essential oils were stabilised in ethanol at a ratio 1:10 (EO: 98% pure ethanol) and stored at −20 °C for future use.

### 4.2. GC and GC-MS Analyses of Essential Oils

Gas chromatographic (GC) analyses were run on a Shimadzu Model 17-A gas chromatograph equipped with a flame ionization detector (FID) and with the operating software Class VP Chromatography Data System version 4.3 (Shimadzu, Kyoto, Japan). The analytical conditions included an SPB-5 capillary column (15 m × 0.10 mm × 0.15 μm) and helium as the carrier gas (1 mL/min). Injection was performed in split mode (1:200) with an injection volume of 1 μL (4% essential oil/CH2Cl2 *v/v*) and with injector and detector temperatures of 250 and 280 °C, respectively. The linear velocity in the column was 19 cm/s. The oven temperature was held at 60 °C for 1 min and then followed a program with an increase from 60 to 280 °C at 10 °C/min and then a hold at 280 °C for 1 min. The percentages of the compounds were determined from their peak areas in the GC-FID profiles.

Gas chromatography–mass spectrometry (GC-MS) was carried out in fast mode on a Shimadzu GC-MS (model GCMS-QP5050A) with the same column and operating conditions used for analytical GC-FID and with the operating software GCMS Solution version 1.02 (Shimadzu). The ionization voltage was 70 eV, the electron multiplier was 900 V, and the ion source temperature was 180 °C. Mass spectra were acquired in the scan mode in the *m/z* range 40–400. The same oil solutions (1 μL) were injected with the split mode (1:96).

### 4.3. Identification of Components of the Essential Oils

The identification of components was based on their GC retention indices (relative to C9-C20 n-alkanes on the SPB-5 column), computer matching of spectral MS data with those from NIST MS libraries [47], comparison of the fragmentation patterns with those reported in the literature [48] and, whenever possible, co-injections with authentic samples.

### 4.4. Cell Culture and EO Treatments

M14 melanoma cells were cultured in RPMI-1640 medium supplemented with 10% foetal calf serum (Euroclone, Pero, Milan, Italy), 2 mM L-glutamine, and 1% penicillin/streptomycin in a fully humidified incubator containing 5% CO_2_ at 37 °C. The identity of the M14 cell line was confirmed and certified by analysing the genetic characteristics of the cell line by PCR-single-locus-technology. Twenty-one independent PCR systems were investigated using the Promega, PowerPlex 21 PCR Kit: Amelogenin, D3S1358, D1S1656, D6S1043, D13S317, Penta E, D16S539, D18S51, D2S1338, CSF1PO, Penta D, TH01, vWA, D21S11, D7S820, D5S818, TPOX, D8S1179, D12S391, D19S433 and FGA. In parallel, positive and negative controls were carried out, yielding correct results. The genetic results were then compared with the online database of the DSMZ. The Eurofins Medigenomix, Forensik GmbH Company (Ebersberg, Germany) performed the analysis.

The cytotoxicity of *C. zeylanicum* EOs was evaluated by analysis of dose–response growth curves of M14-treated cells. *C. zeylanicum* EOs were administered at three different doses (0.1, 1, and 10 μg/mL). Viable cell counts were obtained (by Trypan blue exclusion assays) every day for up to 3 days of growth (after treatment periods of 72 h and 48 h). EO6 displayed a higher cytotoxic effect than the other five EOs tested (Appendix A); therefore, all subsequent experiments were performed with EO6.

Buffy coats (BCs) of hu-PBMCs and MDMs were obtained after centrifugation and separation of whole blood collected from healthy repeat blood donors at the Immunohematology and Transfusion Medicine Unit, Policlinico Umberto I, Sapienza University of Rome, Italy (approved by CNR Ethics with the <Ethical Review ISB Buffy Coats>, Protocol Number 73002/2023). The buffy coats were diluted 1:1 with phosphate-buffered saline (PBS), and mononuclear cells were separated in a Ficoll (Eurobio, Paris, France) gradient. The cells were harvested, washed twice, and plated at a concentration of 2 × 10^6^ cells/mL in culture flasks. The cultures were kept in glutamine-enriched RPMI 1640 medium supplemented with gentamicin (10 mg/L) and 20% foetal calf serum (FCS), and the cells were incubated at 37 °C in a 5% CO_2_/95% air-like atmosphere. After 12 h of adherence, the supernatant was discarded; the attached macrophages were washed twice, detached with cold PBS through gentle scraping, and plated at a concentration of 1 × 10^6^ cells/mL.

Treatment of MDMs with CINN-EO was performed on the seventh day of culture. Human MDMs were treated with three different dilutions of CINN-EO in RPMI medium, D1 (1:1000), D2 (1:10,000) and D3 (1:100,000; corresponding to 10 µg/mL), and we assessed cell viability after 2 h with Trypan blue staining. None of the tested dilutions revealed cytotoxic effects. Untreated MDMs were kept on a separate plate and in a different incubator to avoid any aerosol contamination by volatile CINN-EO secondary metabolites.

### 4.5. Cell Cycle Analysis

After 24 h of treatment with CINN-EO, M14 cells were harvested, washed in 1X PBS, and then fixed in 70% ethanol for at least 1 h at a concentration of 1 × 10^6^ cells/mL. The fixed cells were stained in a solution containing 50 µg/mL PI and 75 KU/mL RNase in 1X PBS for at least 30 min in the dark. A total of 20,000 events per sample were acquired by using a FACSCalibur cytofluorimeter and CellQuest Pro BD software (BD, Franklin Lakes, NJ, USA). The fractions of the cells in the different cell cycle stages were estimated with linear PI histograms by using ModFit software (BD).

### 4.6. ROS Production and Mitochondrial Potential Assays

ROS (reactive oxygen species) generation was measured by FACS using the oxidant-sensitive probe 2′,7′-dichlorodihydrofluorescein (H2DCF-DA). *C. zeylanicum* EO was administered at a dose of 10 µg/mL for 24 h. The treated cells were then harvested and incubated with 10 µM H2DCF-DA in HBSS with 0.1% BSA for 30 min at 37 °C. To generate ROS as a control, we used H_2_O_2_ at a concentration of 5 mM. The cells were then washed in HBSS with 0.1% BSA and analysed with a FACSCalibur cytofluorimeter and CellQuest Pro BD software (BD).

The mitochondrial membrane ΔΦ was measured by FACS using JC-1 staining [49]. After washing in PBS, the cells were incubated with 2.5 mg/mL JC-1 for 20 min at room temperature in the dark. After two washes in PBS, the samples were immediately analysed with a FACSCalibur cytofluorimeter and CellQuest Pro BD software (BD). As a control, we used a depolarised sample treated with the ionophore valinomycin for an additional 15 min after JC-1 staining.

### 4.7. Intracellular Ferrous Iron Measurement with a Trx-puromycin Probe: Puromycin Incorporation Analysis Via In-Cell Immunofluorescence

A trioxolane-derived iron-sensitive probe was recently described by Renslo et al., and ferrous iron pool detection can be achieved with a probe in which Fe(II)-induced trioxolane fragmentation serves to dissociate a FRET pair [15]. Briefly, the authors conjugated puromycin to a previously described 1,2,4-trioxolane scaffold to produce the cell-active probe Trx-Puro-3. Puromycin is incorporated into nascent polypeptides at ribosomes, creating a permanent and dose-dependent mark on cells that can be detected with puromycin-specific antibodies. Notably, the α-amino group of puromycin required for incorporation into peptides is carbamoylated in conjugate 3, so puromycin incorporation from conjugate 3 is precluded before the reaction with Fe(II). As a negative control, the authors prepared the bioisosteric but nonperoxidic dioxolane conjugate 4, which does not react with Fe(II).

M14 cells were exposed to puromycin (Sigma Aldrich, St. Louis, MO, USA) or conjugates thereof (Trx-3-Puro or Trx-4-Puro) at the concentration suggested by Renslo’s laboratory (1 µM, diluted in cell culture medium from 1000 × DMSO stocks) in medium for 4 h. The cells were then harvested, washed with PBS, and fixed in 4% PFA in 1X PBS for 10 min at RT. After washing twice with 1X PBS and once with PBS containing 0.1% Triton X-100, the cells were incubated with the monoclonal antibody anti-puromycin (PMY-2A4; Developmental Studies Hybridoma Bank) (1:500) in 10% FBS in PBS with 0.1% Triton X-100 for 30 min at 37 °C. As a secondary antibody, we used an anti-mouse Alexa Fluor 488 antibody. The cell-associated fluorescence was then analysed with a FACSCalibur cytofluorimeter and CellQuest Pro BD software (BD).

### 4.8. Gene Expression Profiling by qRT-PCR

M14 and human PBMCs and MDMs were drained of medium, and the adherent cells were suspended in ice-cold 4 M guanidium iso-thiocyanate (GTC) lysis solution. Total RNA was extracted as described in previous studies [50,51], analysed in a 1.5% denaturing agarose gel for the absence of degradation, and quantified by UV spectroscopy at 260/280 nm. One microgram of total RNA was reverse-transcribed using random hexamers and SuperScript III Reverse Transcriptase (Invitrogen, Paisley, UK) according to the manufacturer’s instructions. Quantification of PCR products was performed with an ABI PRISM 7500 FAST instrument (Thermo Fisher Scientific, Waltham, MA, USA). Real Master SYBR Green (Thermo Fisher Scientific, MA, USA) was used to produce fluorescent-labelled PCR products, and we monitored the increasing fluorescence during repeated cycles of the amplification reaction. The primer sets for all amplicons were designed using Primer-BLAST software (https://www.ncbi.nlm.nih.gov/tools/primer-blast (accessed on 15 January 2021). For all primers, the following temperature cycling profile was used: 2 min at 50 °C and 2 min at 95 °C followed by 1 min and 30 s at primer-specific annealing temperatures (for all the primer pairs, the annealing T was chosen to be approximately 60 °C (±3 °C) for 40 cycles. GAPDH was used as an internal control because it was shown to be stable with different inductions (primer sequences). The relative level for each gene was calculated using the 2^−ΔΔCt^ method [52]. The sequences for the primers are reported in Appendix A.

### 4.9. Drugs and C. zeylanicum EO Co-administration and Colony Formation Assay

M14 cells were exposed to 10 µg/mL *C. zeylanicum* EO alone or in combination with (Z)-4-Hydroxytamoxifen (Sigma-Aldrich, #H7904; TAM) at a concentration of 0.1 µM in ethanol, paclitaxel (Sigma-Aldrich, #T7402; PTX) at a concentration of 10 nM in DMSO, and dabrafenib (Selleck Chemicals, Houston, TX, USA; #GSK2118436; DAB) at concentrations of 1 nM and 1 µM in DMSO for 24 h. The drug doses employed were chosen by performing dose–response experiments on M14 cells using different doses in each case: TAM (0.1, 1, and 10 µM); PTX (10, 30, and 100 nM); and DAB (0.1, 1, 10, 100, and 1000 nM). The lowest TAM (0.1 µM) and PTX (10 nM) were used in the combination treatment experiments since they did not elicit any toxic effect on M14 cells. One very low and the highest DAB doses were chosen as M14 cells appear to be relatively resistant to this drug (Appendix A). In each case, adherent cells were harvested after treatment and seeded at clonal density (1000 cells/dish) in 35 mm Petri dishes. Fifteen days after seeding, a solution of 2% methylene blue in 95% ethanol was added to the monolayer for at least 30 min. The dishes were then washed with ddH_2_O, and the colonies (at least 50 cells) were counted. The results are expressed as the plating efficiency (percentage of colonies formed from the number of cells seeded). The percentage of cell survival was calculated as the % survival of control cells.

### 4.10. Western Blot Analysis

Cultured cells were washed twice with 1X PBS and then incubated for 1 min in urea buffer (8 M urea, 100 mM NaH_2_PO_4_, and 10 mM Tris pH 8), scraped, harvested and briefly sonicated (10 s). The proteins were subjected to SDS–polyacrylamide gel electrophoresis. The resolved proteins were blotted overnight onto nitrocellulose membranes, which were then blocked in 1X PBS containing 5% non-fat milk for at least 1 h. The blots were incubated with the following primary anti-human antibodies: rabbit polyclonal anti-Cyclin B1 (H433; Santa Cruz Biotechnology, Dallas, TX, USA); mouse monoclonal anti-P53 (DO-7; Dako, Glostrup, Hovedstaden, Denmark); mouse monoclonal anti-Transferrin (clone #507506; R&D Systems, Minneapolis, MN, USA); mouse monoclonal anti-HMOX1 (sc-136960; Santa Cruz Biotechnology, Santa Cruz, TX, USA); mouse monoclonal anti-GAPDH (6C5; Millipore, Burlington, MA, USA); and mouse monoclonal anti-HSP 72/73 (Ab1-W27; Calbiochem, San Diego, CA, USA). The membranes were then incubated for 45 min with the appropriate secondary antibody: donkey anti-rabbit IRdye800 (LI-COR Biotechnology, Lincoln, NE, USA) or donkey anti-mouse IRdye800 (LI-COR). The membranes were then analysed with a LI-COR Odyssey Infrared Image System in the 800 nm channel.

### 4.11. Effect of the HMOX1 Inhibitor SnPPIX on M14 Cell Proliferation upon CINN-EO Treatment

M14 cell metabolic activity and proliferation were determined with CellTiter 96^®^ Aqueous One Solution Reagent from a cell proliferation colorimetric assay (Promega, Madison, WI, USA) upon CINN-EO administration and with or without tin protoporphyrin IX (SnPPIX), a well-characterised HMOX-1 enzymatic inhibitor. Melanoma cells were incubated with 10 µg/mL CINN-EO alone or in combination with 5 µM SnPPIX for 24 h. As a control, cells were also incubated with 5 µM SnPPIX alone. After treatment, the cells were incubated with 100 mL/mL MTS, the tetrazolium compound 3-(4,5-dimethylthiazol-2-yl)-5-(3-carboxymethoxyphenyl)-2-(4-sulfophenyl)-2H-tetrazolium on which this assay is based, and an electron-coupling reagent (phenazine methosulfate) at 37 °C for approximately 1 h. Metabolically active cells reduced MTS into a soluble formazan product, the absorbance of which was measured at 490 nm in growth medium. The background absorbance of the medium-only control was subtracted from the absorbance of the collected samples.

### 4.12. Statistical Analysis

Statistical analysis was performed using GraphPad Prism 5 software (GraphPad software, Inc., La Jolla, CA, USA). One-way ANOVA and Bonferroni post-test was used for groups of data. Mann–Whitney or Unpaired student’s t test was used for comparison of pairs of data.

## 5. Conclusions

In conclusion, our data show that the cytotoxic effect of a *C. zeylanicum* EO on the M14 metastatic melanoma cell line is associated with increases in ROS and Fe(II) and a reversible accumulation of cells in the G2/M cell cycle phases. We also show that CINN-EO, by inducing an incomplete stress response, can enhance conventional drug antitumour effects. In the light of the therapeutic effects described (as demonstrated for dabrafenib), a therapeutic strategy based on the administration of *C. zeylanicum* EO in combination with one or more anticancer drugs could allow the effectiveness of the very same drugs to be increased at the standard doses or for the needed drug doses to be decreased. At the same time, it provides a more effective solution in the treatment of metastatic melanoma than conventional therapeutic protocols. In fact, the administration of the CINN-EO in association with hydroxytamoxifen is a possible alternative treatment option for patients who do not have the BRAF V600 mutation and cannot be treated with BRAF inhibitors.

## 6. Patents

Patent Application Number: PCT/I2022/054196 (date of receipt, 6 May 2022; receiving office, International Bureau of WIPO).

## Figures and Tables

**Figure 1 ijms-24-05698-f001:**
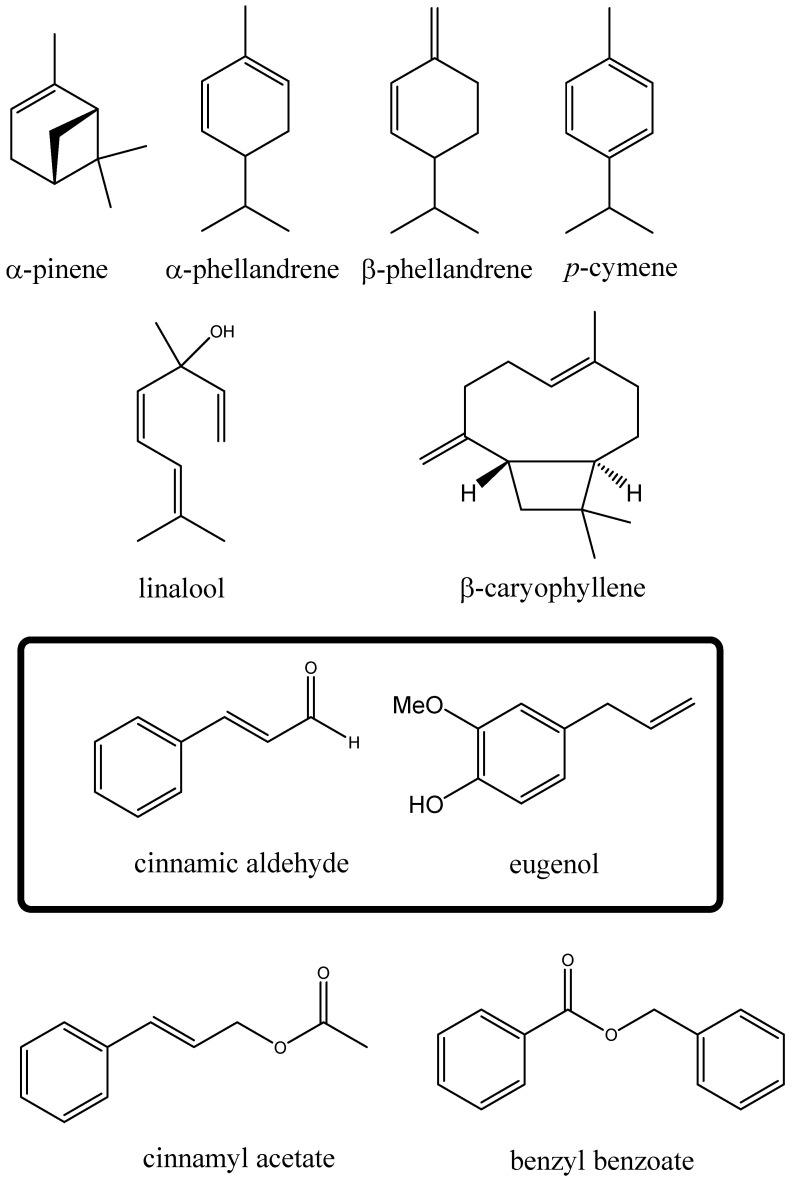
Molecular formula of the main components of cinnamon essential oils.

**Figure 2 ijms-24-05698-f002:**
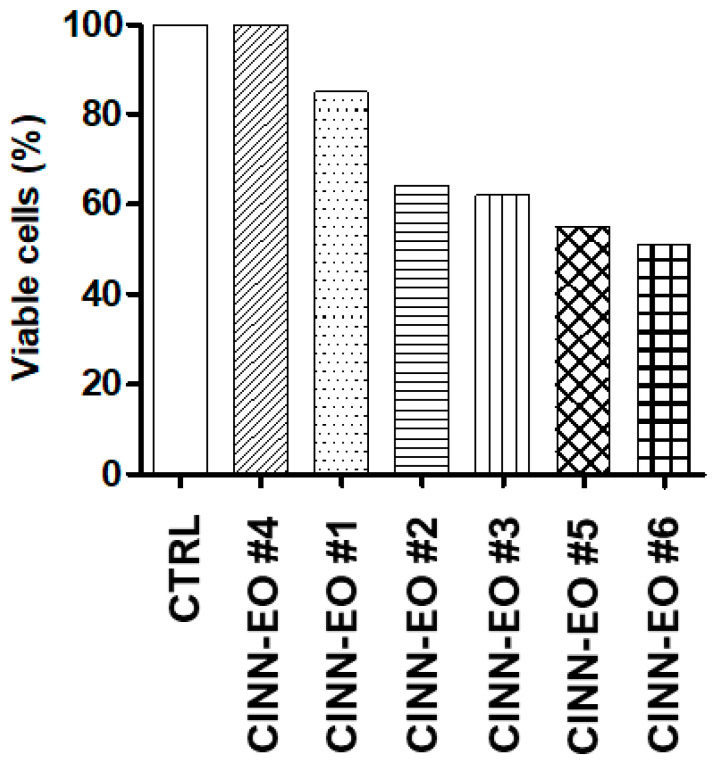
Dose–response effect of six different *C. zeylanicum* essential oils (CINN-EOs). All the CINN-EOs were used at the concentration of 10 µg/mL on M14 cells for 48 h. The number of viable cells after each treatment is reported as a percentage of the untreated control cells.

**Figure 3 ijms-24-05698-f003:**
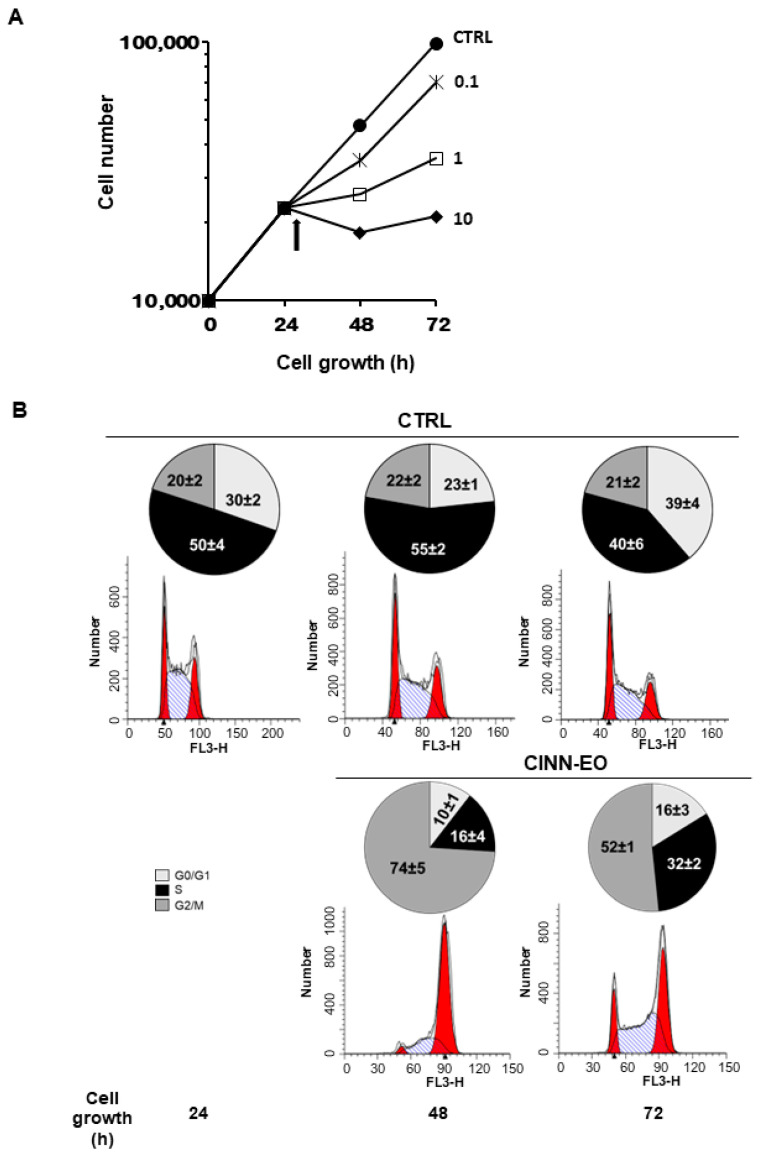
Anti-proliferative effect of CINN-EO on M14 melanoma cells. (**A**) Dose–response curves for CINN-EO at the indicated oil concentrations (µg/mL). CTRL, untreated cells. The values are the mean ± SD of at least three experiments. The standard deviations, when not evident, are comprised in symbols. The arrow indicates the start of CINN-EO treatment. (**B**) Effect of CINN-EO on the cell-cycle phase distribution of the M14 melanoma cell line. Cells were exposed to 10 µg/mL CINN-EO for the indicated times. Untreated (CTRL) and treated cells were then stained with PI and analysed for DNA content by flow cytometry. The percentages ± SDs of cells in the G0/G1 (light grey), S (black) and G2/M (dark grey) phases of the cell cycle were estimated from each histogram with ModFit software. The histograms shown are representative ModFit analyses for each condition from at least three different experiments with similar results. The number reported in each panel represents the time of treatment (h).

**Figure 4 ijms-24-05698-f004:**
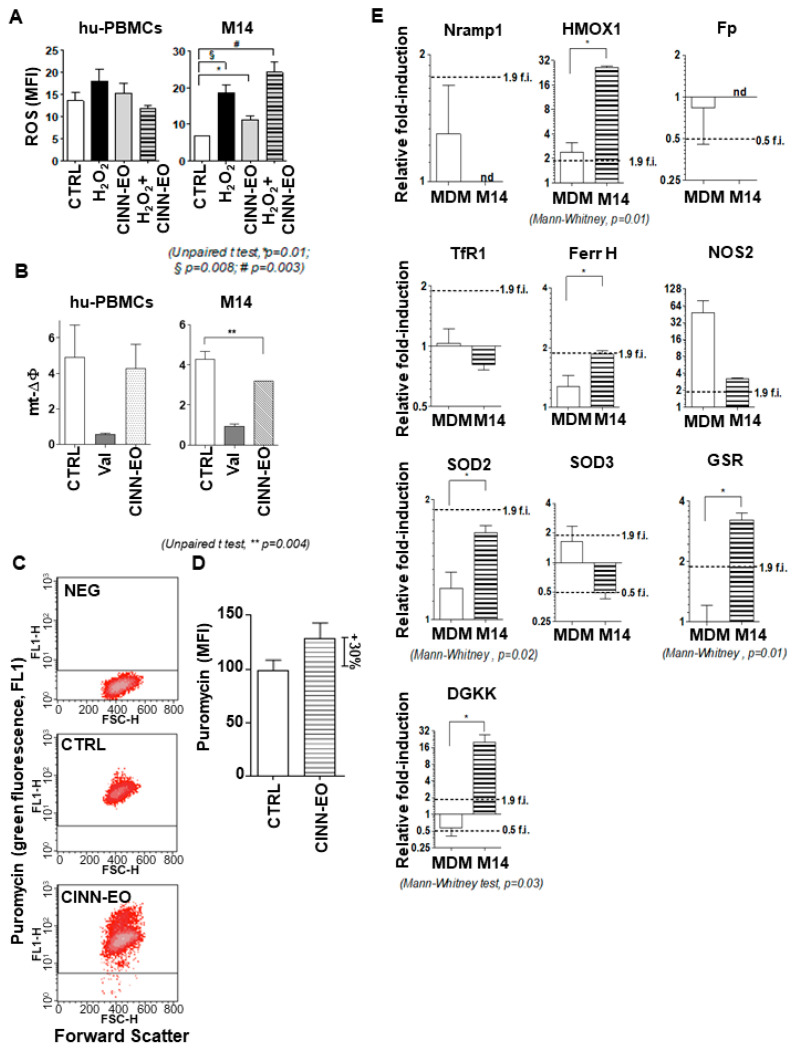
CINN-EO affects intracellular oxidative and iron metabolism. (**A**) ROS production in M14 cells and hu-PBMCs. (**B**) Mitochondrial membrane potential (mt-ΔΦ) in M14 cells and hu-PBMCs. (**C**) Representative cytograms of puromycin induction after CINN-EO treatment in M14 cells. CTRL, untreated cells; CINN-EO, treated cells. (**D**) Puromycin mean fluorescence intensity (MFI), as evaluated by FACS and normalised to the cell protein content, indicating the intracellular iron(II) amount after CINN-EO treatment. The results are the average of three independent experiments. (**E**) Gene expression in MDMs and M14 cells upon CINN-EO treatment (* *p* < 0.05; nd, not detected). Relative fold-induction were calculated versus control MDM.

**Figure 5 ijms-24-05698-f005:**
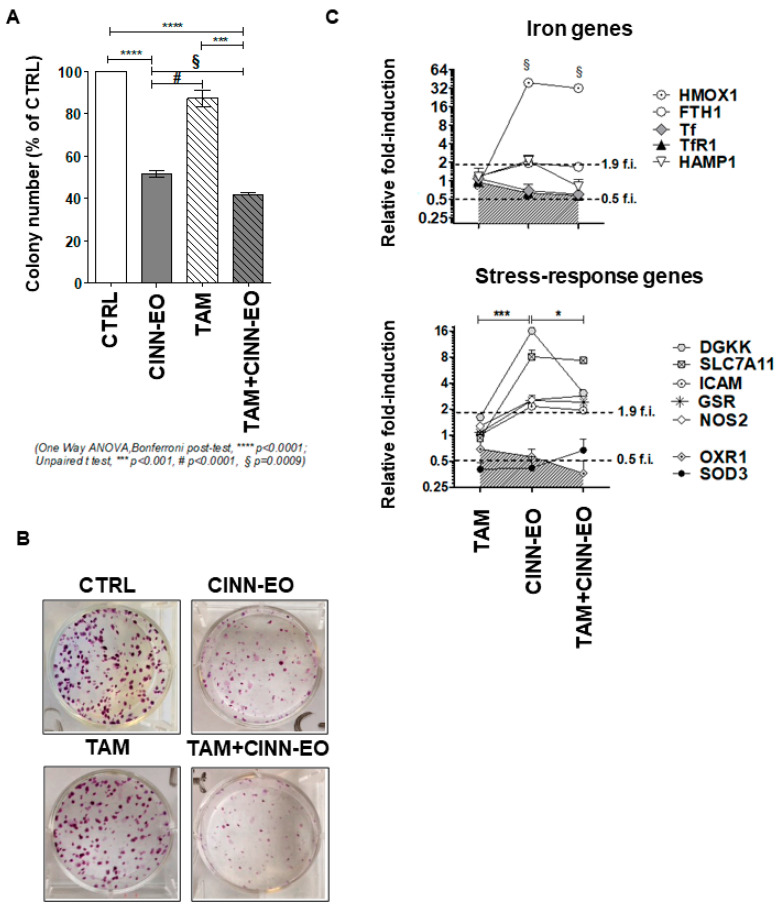
Effects of the combination of CINN-EO with the drug tamoxifen on M14 cell survival. (**A**) Colony formation assay on M14 cells exposed to CINN-EO alone or in combination with tamoxifen for 24 h. The cell survival percentages were calculated as the percentages of colonies formed in each sample versus the percentage of colonies formed in controls. (**B**) Images of the stained colonies obtained in each sample. (**C**) Expression of the indicated genes after CINN-EO ± TAM treatment. Relative fold-induction is calculated versus untreated M14 cells. * *p* < 0.05; *** *p* < 0.0001; § *p* < 0.0001.

**Figure 6 ijms-24-05698-f006:**
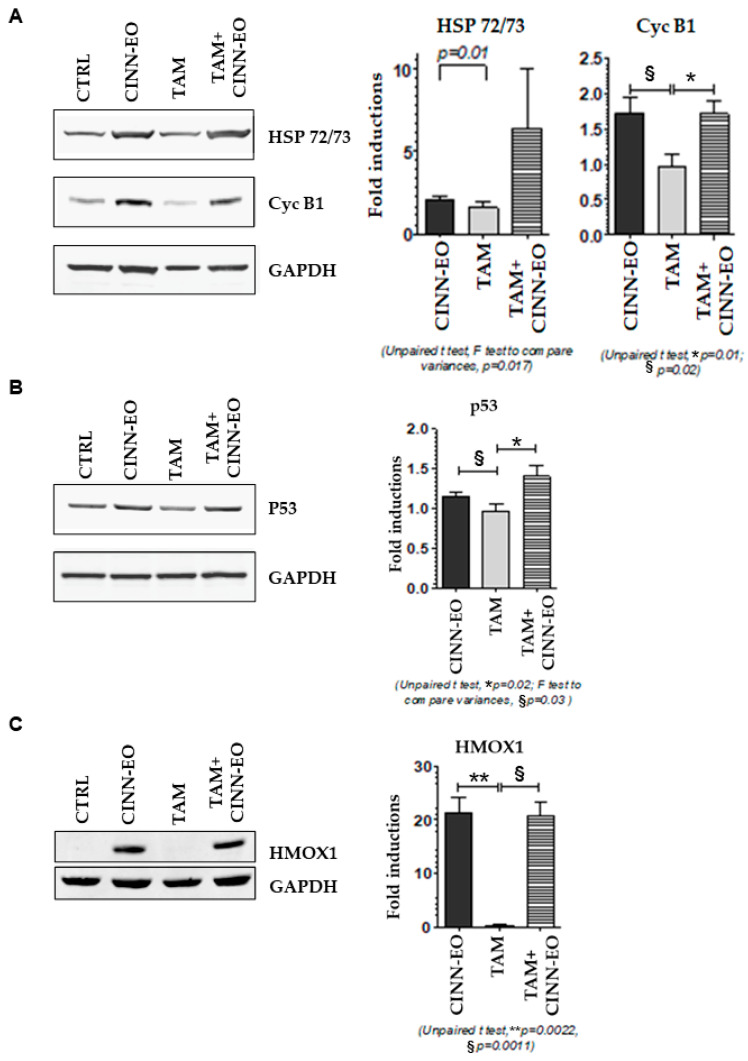
Effects of CINN-EO alone or in combination with TAM on the expression of the indicated proteins. (**A**) HSP72/73 (upper) and Cyclin B1 (middle) protein expression fold-induction vs. Ctrl (right); corresponding representative blots are shown (left). (**B**) p53 protein expression fold-induction vs. Ctrl (right) and corresponding representative blots (left). (**C**) HMOX1 protein expression fold-induction vs. Ctrl (right) and corresponding representative blots (left). GAPDH expression was used as a protein loading control in each blot. Fold-inductions are calculated for each protein versus the untreated CTRL.

**Figure 7 ijms-24-05698-f007:**
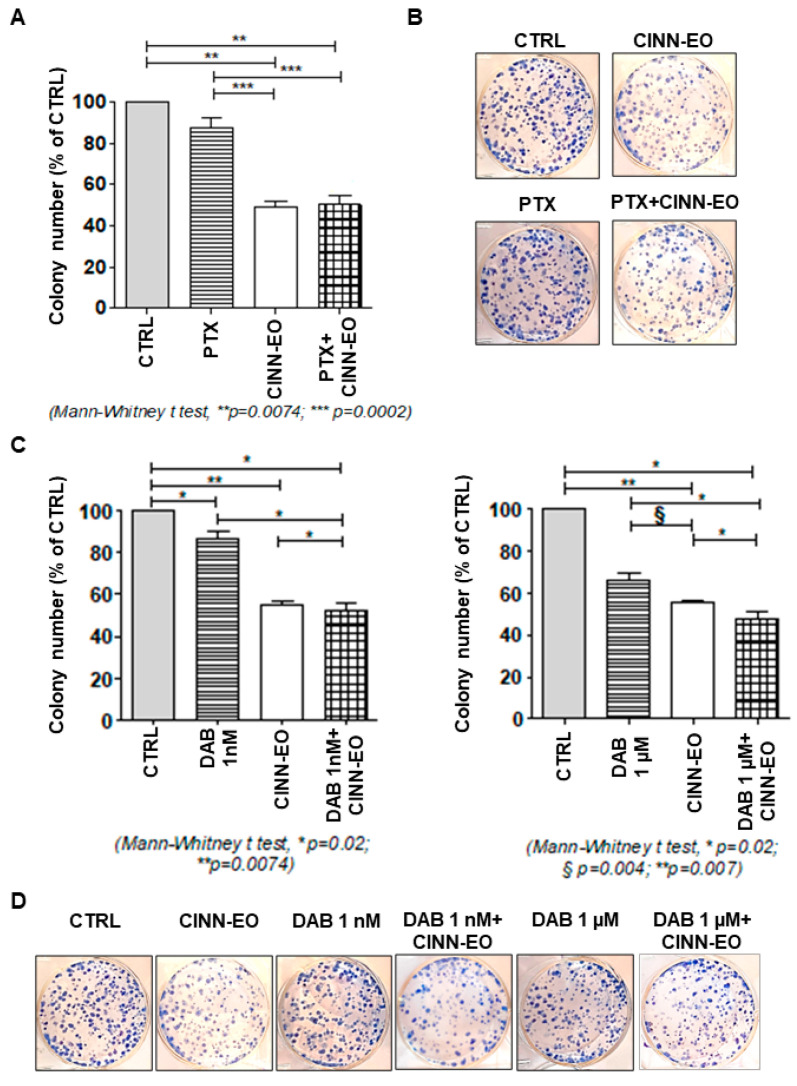
Effects of the combination of CINN-EO with the drugs paclitaxel (PTX) and dabrafenib (DAB) on M14 cell survival. (**A**) Colony formation assay on M14 cells exposed to CINN-EO alone or in combination with PTX for 24 h. The cell survival percentages were calculated as the percentages of colonies formed in each sample versus the percentage of colonies formed in untreated controls. (**B**) Images of the stained colonies obtained in each sample. (**C**) Colony formation assay on M14 cells exposed to CINN-EO alone or in combination with DAB 1nM (left panel) or 1µM (right panel) for 24 h. The cell survival percentages were calculated as the percentages of colonies formed in each sample versus the percentage of colonies formed in untreated controls. (**D**) Images of the stained colonies obtained in each sample.

**Figure 8 ijms-24-05698-f008:**
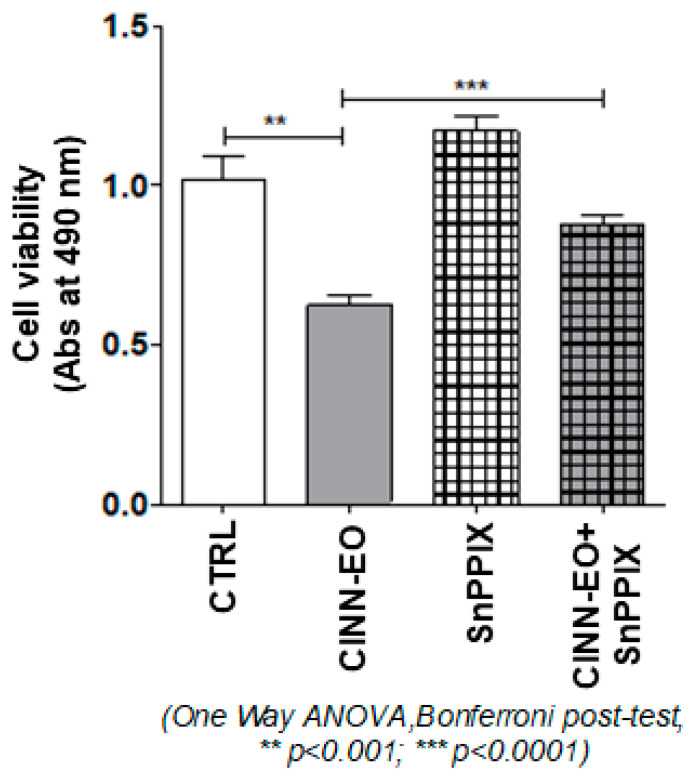
SnPPIX rescues the inhibition of cell proliferation induced by CINN-EO on M14 cells. Results of an MTS assay on M14 cells exposed to CINN-EO alone or in combination with the HMOX1 inhibitor SnPPIX (5 µM) for 24 h. The values are expressed as the mean ± SD from three independent experiments.

**Table 1 ijms-24-05698-t001:** Cinnamon essential oil (CINN-EO) samples.

Samples	Label
SAMPLE 1—commercial from ERBA VITA GROUP S.p.A., Chiesanuova (RSM), batch # 34514	EO1
SAMPLE 2–commercial from ERBA VITA GROUP S.p.A., batch # 31913	EO2
SAMPLE 3—commercial from ZUCCARI, Trento (TN), batch # 15	EO3
SAMPLE 4—commercial from RAO ERBE S.r.l., Valverde (CT)	EO4
SAMPLE 5—commercial from GALENO S.r.l., Comeana (PO), batch # 02193319	EO5
SAMPLE 6—commercial from GALENO S.r.l., Comeana (PO), batch # 02202006	EO6

## Data Availability

Not applicable.

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
