# Peer review of "Cinnamomum zeylanicum Blume Essential Oil Inhibits Metastatic Melanoma Cell Proliferation by Triggering an Incomplete Tumour Cell Stress Response"

_ijms, 2023, doi:10.3390/ijms24065698_

Round 1

Reviewer 1 Report

Comments attached

Author Response

Point-by-point answers

Reviewer 1

Comments on Manuscript ID ijms-2113722

Title Cinnamomum zeylanicum Blume essential oil inhibits metastatic melanoma cell proliferation by triggering an incomplete tumour cell stress response.

Authors Giulia Cappelli et al

Comments:

There is a prime need to explore the possibility of the alternative novel therapeutic approaches by using medicinal plant extracts in the form of natural products to overcome the cancer in the form of various types and stages of the melanoma. The authors have systematically to explore the potential role of Cinnamomum family members of the medicinal plant which might provide the promising role to cure and or prevent the melanoma form of cancer. The authors are expected to address the following concerns.

  1. What is the criteria of the selection of 6 samples of Cinnamon essential oil (CINN-EO) samples, please explain the matching rationale behind the selection.

A: We have chosen only CINN-EOs, which could be edible, and which were obtained from plants grown in organic farming

  1. The authors stated the oil sample EO4 is eugenol is clove oil, which is aromatic oil extracted from cloves what was the inclusive criteria of this oil and how is corelating with the 5 samples of CINN-EO.

A: We never said that EO4 was obtained by Clove, or Syzygium aromaticum. We only noted that, in the composition of this EO, Eugenol was higher than in all the other 5 CINN-EOs.

As specified in the new Table 1, all the 6 EOs were derived from C. zeylanicum bark.

  1. The author placed compositions of the cinnamon essential oil samples in the table 1, the details list of this table should be placed in the supplement section of the manuscript instead of at the main text.

A: Following the Reviewer’s comment we moved table 1 in the supplementary material and changed numbering accordingly.

  1. Authors team need to establish the synergetic correlation of the compositions of the selected group of 6 oils with their role in the therapeutic efficacy.

A: We thank the Reviewer for the suggestion, but the issue raised is very complex to address. Who work with essential oils know that they are well known to be rather complex matrices with tens or even hundreds of components. And this is our case, where each of the 6 EOs taken into consideration contains just under a hundred components. The experimental effort needed to establish the synergistic correlation of the compositions of the selected group of 6 oils with their role in therapeutic efficacy is very high, and it is undoubtedly impossible to address the problem in a concrete way, unfortunately we do not see the conditions. Furthermore, we do not see the added value that an experiment of this extent can bring to our work. We think that knowing which components of our best tested oil are most effective in the observed therapeutic effect would not add much more to the meaning we have given to our work. Our main goal was to demonstrate that Cinnamomun zeylanicum essential oil has an effect in inhibiting tumor growth and enhancing the effect of conventional anticancer drugs to be able to use low doses of chemotherapy and this we think has been demonstrated in the work we have submitted.

In conclusion, in our opinion, without the supporting of experimental data, we should only make mere speculations on concepts such as synergism and antagonism, which, would not bring any improvement to the work.

We hope the Reviewer agrees with our view on the requested change.

  1. The authors selected the EO6 based on the highest cytotoxicity capacity and referred as CINN-EO, please also provide the Cytotoxicity level of all 6 samples in the main text of the manuscript in the Result section 2.2.

A: Following the Reviewer’s comment, we added the cytotoxicity data of all 6 EOs as a figure in the main text, Figure 2 of the revised version.

  1. In the result section 2.3 line 287, the author need to replace the word wanted with the scientific terminology evaluate.

  1. Following the Reviewer’s comment we changed the term “wanted to make sure” with the more appropriate term “ascertained". Page 10, line 352 of the revised version.

Reviewer 2 Report

According to the authors, the main objective of this study was to evaluate the effect of C. zeylanicum essential oil (CINN-EO) on human metastatic melanoma, particularly by assessing the stress response, which included iron metabolism (ferroptosis) and also gene expression of the stress response. Even though interesting data were presented there are another couple of factors that should be concerned.

In the Abstract section:

1) What cell type(s) was (were) used in the study? Including control cells.

In the Introduction section:

2) Please, provide adequate references for the first paragraph.

3) Since the study focused on oxidative stress, it is important to define it.

4) In order to be clearer, the authors need to present the objective(s) of the study in the last paragraph of this section. In addition, I also suggest a revision and moving the last paragraph of this section, which described some of the main results obtained in this study, to the first paragraph of the "Discussion" section.

In the Material and Methods section:

3) Please, describe the Statistical analysis.

4) In subsection "4.4. Cell culture and treatments", it is necessary to describe the origin of TAM, and also there is a lack of information regarding PTX and DAB.

5) Since the study used PMBC, which was obtained from healthy donors, there is important to present the Ethical approval number.

In the Results section

6) Please, provide the supplementary S1 figure.

7) Please, provide the cytotoxic effect of CINN-EO treatment in the PBMC and MDMs.

8) Please, revise the information presented in the sentence on page 11, lines 208-209.

9) Although the authors point out that "Where administration of PTX alone elicited an inhibitory effect on M14 CFA of less than 10%, administration of CINN-EO in association with PTX decreased M14 cell survival by more than 60% (p=0.0002) (Figure 6 A).", page 13, lines 265-267, in accordance with this figure, the administration of PTX alone or in combination with CINN-EO did not show any effect. Based on these data, the antimitotic effect of PTX was not verified. Please, clarify this point.

In the Discussion section:

10) As previously mentioned, I suggest introducing this section describing the most important results obtained in the present study.

11) On page 15, lines 313-314, the authors affirmed that "The increased ROS production was responsible for the inhibition of cell growth observed following CINN-EO exposure in M14 cells." However, since it was not provided results obtained in experiments in which antioxidant agents were used, which inhibit the ROS effect, this affirmation is fully true.

12) On page 16, lines 341-343, the authors affirmed that "We showed that co-administration of CINN-EO and TAM or PTX or DAB had an effect superior to that of the two individual treatments on M14 survival and could allow the use of the drugs at doses lower than the one employed in current protocols." However, as previously mentioned, the use of PTX did not affect the M14 survival, and also the reduction of these cells' survival was observed in the CINN-EO alone. Please, revise this sentence.

13) On page 17, lines 419-420, the authors affirmed that "Nevertheless, in our experiments, CINN-EO did not display cytotoxic effects towards human MDMs." Where were these data shown?

14) What are the limitations of the study?

In the Conclusion section:

15) On page 21, lines 587-593, the authors affirmed that "In the light of the therapeutic effects described, a therapeutic strategy based on the administration of C. zeylanicum EO in combination with one or more anti-cancer drugs could allow decreasing the needed drug doses (as demonstrated for PTX and DAB). At the same time, it provides a more effective solution in the treatment of metastatic melanoma than conventional therapeutic approaches. In addition, the administration of the CINN-EO in association with TAM, it is a possible alternative treatment option for patients who do not have the BRAF V600 mutation and cannot be treated with DAB." Despite some results presented being very interesting and can drive further studies, it is reckless to affirm these points of view since this is an in vitro study, mainly by using only one melanoma cell line. Thus, revise the "Conclusion" section.

Author Response

Reviewer 2

Comments and Suggestions for Authors

According to the authors, the main objective of this study was to evaluate the effect of C. zeylanicum essential oil (CINN-EO) on human metastatic melanoma, particularly by assessing the stress response, which included iron metabolism (ferroptosis) and also gene expression of the stress response. Even though interesting data were presented there are another couple of factors that should be concerned.

In the Abstract section:

1) What cell type(s) was (were) used in the study? Including control cells.

A: Following the Reviewer’s suggestion, we added in the Abstract the specification of the tumoral, and healthy cells used in the study. We also added specification about the authenticity of the M14 melanoma cells in Mat&Met section.

In the Introduction section:

2) Please, provide adequate references for the first paragraph.

A: Following the Reviewer’s comment we added some new references in the Introduction section.

3) Since the study focused on oxidative stress, it is important to define it.

A: Following the Reviewer’s comment we integrated the first part of the Introduction in the revised version.

4) In order to be clearer, the authors need to present the objective(s) of the study in the last paragraph of this section. In addition, I also suggest a revision and moving the last paragraph of this section, which described some of the main results obtained in this study, to the first paragraph of the "Discussion" section.

A: Following the Reviewer’s comment we integrated the Introduction indicating our main objectives in the revised version. We also moved the last paragraph of the Introduction at the beginning of the Discussion.

 In the Material and Methods section:

3) Please, describe the Statistical analysis.

A: We apologize for not including the paragraph on statistical analysis. It was added as paragraph 4.12 in the revised version.

4) In subsection "4.4. Cell culture and treatments", it is necessary to describe the origin of TAM, and also there is a lack of information regarding PTX and DAB.

A: According to the Reviewer’s comment we added information about the three drugs employed in our experiments. Information have been added in section 4.9.

5) Since the study used PMBC, which was obtained from healthy donors, there is important to present the Ethical approval number.

A: According to the Reviewer’s comment we added the ethical approval protocol number in the revised version of the manuscript. Page 17, second paragraph.

In the Results section

6) Please, provide the supplementary S1 figure.

A: We apologize because we realized that we have not uploaded the supplementary figure file information. In the submission of the revised version, we did it. However, since Reviewer #1 asked to insert in the main text (Result section 2.2) the cytotoxicity level of all 6 oil samples (we described previously in supplementary Figure S1), these data have been now added as Figure 2 in the revised version.

7) Please, provide the cytotoxic effect of CINN-EO treatment in the PBMC and MDMs.

A: According to the Reviewer’s request, we added the cytotoxicity in the human PBMCs and MDMs as supplementary Figure S1in the revised manuscript.

8) Please, revise the information presented in the sentence on page 11, lines 208-209.

A: Hoping we correctly interpreted the Reviewer’s comment we modified the sentence indicated, page 11, line 375 of the revised version.

9) Although the authors point out that "Where administration of PTX alone elicited an inhibitory effect on M14 CFA of less than 10%, administration of CINN-EO in association with PTX decreased M14 cell survival by more than 60% (p=0.0002) (Figure 6 A).", page 13, lines 265-267, in accordance with this figure, the administration of PTX alone or in combination with CINN-EO did not show any effect. Based on these data, the antimitotic effect of PTX was not verified. Please, clarify this point.

A: We apologize for confusing the Reviewer and agree that we have described the results relative to Figure 6A (Figure 7A of the revised version) incorrectly. According to the Reviewer's comment, we have changed this description (page 13, lines 442-447 of the revised version).

In the Discussion section:

10) As previously mentioned, I suggest introducing this section describing the most important results obtained in the present study.

A: According to the Reviewer’s comment we added the description of the most important results at the beginning of this section (see also comment 4).

11) On page 15, lines 313-314, the authors affirmed that "The increased ROS production was responsible for the inhibition of cell growth observed following CINN-EO exposure in M14 cells." However, since it was not provided results obtained in experiments in which antioxidant agents were used, which inhibit the ROS effect, this affirmation is  fully true.

A: Maybe the reviewer meant that “…this affirmation is NOT fully true”. Interpreting his thought and in agreement with him we modified the indicated sentence. Page 16, lines 526-527 of the revised version.

12) On page 16, lines 341-343, the authors affirmed that "We showed that co-administration of CINN-EO and TAM or PTX or DAB had an effect superior to that of the two individual treatments on M14 survival and could allow the use of the drugs at doses lower than the one employed in current protocols." However, as previously mentioned, the use of PTX did not affect the M14 survival, and also the reduction of these cells' survival was observed in the CINN-EO alone. Please, revise this sentence.

A: In agreement with the Reviewer, we changed the sentence accordingly. Page 16, line 555 of the revised version.

13) On page 17, lines 419-420, the authors affirmed that "Nevertheless, in our experiments, CINN-EO did not display cytotoxic effects towards human MDMs." Where were these data shown?

A: According to the Reviewer’s request, we inserted the cytotoxicity data as Figure S1 of the revised version (see  point 7 above)

14) What are the limitations of the study?

A: For sure, this study has some limitations. First, it is the limited number of melanoma cell lines on which we have tested the CINN-EOs. Although studying the effect of CINN-EOs on a larger number of different melanoma cell lines could have provided a broader scenario of the general effect of EO on this tumor type, to simplify we choose to focus our study on a single melanoma cell line which recapitulated the main malignant parameters of melanoma, i.e. metastatic capacity, the presence of the BRAF V600E mutation, the presence of other genetic alterations such as low PTEN expression and PI3K/Akt pathway activation, as well as the relative resistance to some drugs such as tamoxifen and dabrafenib.

The second limitation that we note in our work is that we did not perform the study in animal models in vivo or even in patients. For this, we can say that although an in vivo study would certainly allow us to evaluate the real translation of our experimental data in the clinic, however, to perform the in vivo study it is first necessary to develop an EO delivery system in an organism. This will allow the administration of the maximum doses of EO allowed by the EU regulatory authority (10 May 2011, EMA/HMPC/246773/2009, Committee on Herbal Medicinal Products (HMPC), Assessment report on Cinnamomum verum J. S. Presl, cortex and corticis aetheroleum.) with a reasonable expected improvement of the therapeutic effects on patients. We think that this could be the objective of a next study.

In the Conclusion section:

15) On page 21, lines 587-593, the authors affirmed that "In the light of the therapeutic effects described, a therapeutic strategy based on the administration of C. zeylanicum EO in combination with one or more anti-cancer drugs could allow decreasing the needed drug doses (as demonstrated for PTX and DAB). At the same time, it provides a more effective solution in the treatment of metastatic melanoma than conventional therapeutic approaches. In addition, the administration of the CINN-EO in association with TAM, it is a possible alternative treatment option for patients who do not have the BRAF V600 mutation and cannot be treated with DAB." Despite some results presented being very interesting and can drive further studies, it is reckless to affirm these points of view since this is an in vitro study, mainly by using only one melanoma cell line. Thus, revise the "Conclusion" section.

A:  According to the Reviewer’s comment, we modified the sentence in the conclusions section. Page 21-22, lines 855-867 of the revised version.

Round 2

Reviewer 2 Report

Based on the fact that the authors not only responded to the questions raised adequately, and also incorporated all the suggestions, which lead to the significant improvement of the meaning of the study, I agree with the publication of the study.